# In Vivo and In Vitro Cartilage Differentiation from Embryonic Epicardial Progenitor Cells

**DOI:** 10.3390/ijms23073614

**Published:** 2022-03-25

**Authors:** Paul Palmquist-Gomes, Ernesto Marín-Sedeño, Adrián Ruiz-Villalba, Gustavo Adolfo Rico-Llanos, José María Pérez-Pomares, Juan Antonio Guadix

**Affiliations:** 1Department of Animal Biology, Faculty of Sciences, Campus de Teatinos s/n, Instituto Malagueño de Biomedicina (IBIMA), University of Málaga, 29080 Malaga, Spain; paul.palmquist@institutimagine.org (P.P.-G.); emarinse@uma.es (E.M.-S.); adruiz@uma.es (A.R.-V.); 2Centro Andaluz de Nanomedicina y Biotecnología (BIONAND), Universidad de Malaga, c/Severo Ochoa 25, Campanillas, Junta de Andalucía, 29590 Malaga, Spain; 3Networking Research Center on Bioengineering, Biomaterials and Nanomedicine, CIBER-BBN, 28029 Malaga, Spain; grico@uma.es; 4Department of Cell Biology, Genetics and Physiology, IBIMA, University of Malaga, 29016 Malaga, Spain

**Keywords:** epicardium, proepicardium, epicardial-derived cells, differentiation, cartilage, chondrocytes, avian embryo, quail to chick chimeras, cardiac regeneration

## Abstract

The presence of cartilage tissue in the embryonic and adult hearts of different vertebrate species is a well-recorded fact. However, while the embryonic neural crest has been historically considered as the main source of cardiac cartilage, recently reported results on the wide connective potential of epicardial lineage cells suggest they could also differentiate into chondrocytes. In this work, we describe the formation of cardiac cartilage clusters from proepicardial cells, both in vivo and in vitro. Our findings report, for the first time, cartilage formation from epicardial progenitor cells, and strongly support the concept of proepicardial cells as multipotent connective progenitors. These results are relevant to our understanding of cardiac cell complexity and the responses of cardiac connective tissues to pathologic stimuli.

## 1. Introduction

Distinct, discrete cartilage tissue clusters have been described in different regions of the adult vertebrate heart. These cartilage clusters often appear in the semilunar valves located at the base of the aortic and pulmonary trunks and adjacent in adjacent tissues of reptiles [1], avians [2,3] and mammals [4]. Indeed, cartilage differentiation is initiated by the condensation of mesenchymal, connective tissue under the instructive signalling provided by key growth factors like BMPs and FGFs and the master regulation of the key transcription factors *Sox9* [5]. This differentiation process results in the active synthesis of a characteristic extracellular matrix (ECM) enriched in collagen II, chondroitin sulphate, hyaluronic acid and several proteoglycans [6].

So far, the only accepted source of cartilage in the heart is the cardiac neural crest [2], a multipotent ectomesenchyme whose cellular cardiac derivatives are mostly limited to the aorto-pulmonary septum, the developing pulmonary and aortic valves and surrounding tissues [7,8,9,10]. However, recent reports have revealed that the embryonic epicardium, i.e., the tissue layer that covers the cardiac muscle (myocardium), also makes a significant contribution to various cardiac connective tissues. The epicardium develops from the proepicardium, a mass of coelomic progenitors located at the venous pole of the embryonic heart. Proepicardial cells attach to and spread over the myocardium to form the primitive epicardial epithelium. Then, the epicardium undergoes an epithelial-to-mesenchymal transition to give rise to a heterogeneous population of epicardium-derived cells (EPDCs) that invade the cardiac interstitium and progressively differentiate into various cell types, including endothelial and smooth muscle cells and cardiac fibroblasts [11,12,13]. The wide mesodermal differentiation potential of the epicardium prompted us to evaluate its ability to differentiate into chondrocytes. In order to do so, we have combined in vivo cell tracing experiments in avian embryos (quail-to-chick proepicardial chimeras) with in vitro differentiation of avian proepicardial cells into chondrocytes. Our results indicate that (pro)epicardial-derived cells have the potential to differentiate into chondrocytes and suggest that at least part of the cartilaginous nodes found in the aortic ring-left cardiac outflow tract domain derive from the epicardial lineage.

## 2. Results

### 2.1. Chondrocyte Clusters Are Present in the Chick Heart

Cartilage-like tissue was observed in the forming proximal outflow tract region of the perinatal chick heart, including the aorta and pulmonary trunks (HH44-46), close to the commissures of semilunar valve leaflets (Figure 1A,B and Appendix A). Near these valvular chondrocyte clusters, additional masses of cartilage-like mesenchymal condensations were identified in a low percentage of embryos (3/20); such clusters were located between the insertion of the aortic ring, the left ventricular outflow tract myocardium, and the left atrioventricular sulcus [14] (Figure 1C and Appendix A, from here onwards LOT). Both chondrocyte clusters were positive for classic cartilage alcian blue and safranin red staining, as well as immunoreactive to collagen type II-alpha 1 (COL2A1) antibody (Figure 1D–I). Interestingly, the embryonic aortic cartilage shows higher intensity of Safranin red and COL2A1 staining than the LOT one (Figure 1E–I). A detailed study of adult chick hearts revealed these chondrocytes accumulated in the same anatomical region described for chick embryos (Figure 1J–P and Appendix A). In these adult hearts, cell density was also significantly higher in aortic and pulmonary trunk chondrocyte clusters than in LOT ones (*p* < 0.01 and *p* < 0.05, respectively) (Figure 1Q and Appendix A).

### 2.2. LOT Cartilage Derives from the (Pro)Epicardium In Vivo

In order to determine the potential contribution of (pro)epicardial-derived cells to any of these chondrocyte clusters, quail-to-chick proepicardial (PE) chimeras were constructed in the avian heart (Figure 2A). In these chimeras, quail proepicardial derivatives were traced in the chick embryonic heart by using the quail specific pan-nuclear QCPN antibody (Figure 2B–C’). These lineage tracing studies are based in the high homology between chick and quail embryonic tissues, including the PE. In accordance, we observed that the aortic root of quail and chick embryos are anatomically comparable (Appendix A).

At perinatal stages, quail proepicardial derived cells (QCPN^+^) were not found in the aortic or the pulmonary valve chondrocyte clusters (Figure 2B,B’ and Appendix A). However, when present, the LOT chondrocytes cluster contained high numbers of QCPN immunoreactive cells (Figure 2C,D and Appendix A). Finally, to test whether the alteration of tissue homeostasis affects epicardial differentiation into chondrocytes in vivo, quail-to-chick proepicardial (PE) chimeras were constructed over cryodamaged chick embryo hosts (Figure 2E). Cryoinjury disrupts ventricular myocardial wall continuity (Figure 2F); the damage site contained large numbers of non-myocardial QCPN^+^ cells (Figure 2G). Seven days after cryoinjury (HH 35-36 stage), no chondrocyte clusters were found in the injured region (*n* = 8) (Figure 2F,G). However, at later stages (16 days after cryoinjury; HH45-46 stage), QCPN^+^ chondrocyte clusters were observed at the wound area in 13% (2/15) of chick damaged hearts (Figure 2H–K). The cartilaginous nature of these cell clusters was confirmed by Safranin red, Alcian blue, and SOX9 nuclear staining (Figure 2I–K); QCPN immunorreactivity was used to reveal the proepicardial origin of cells (Figure 2J).

### 2.3. Proepicardium-Derived Cells Differentiate into Chondrocytes In Vitro

To confirm that proepicardial cells can differentiate into chondrocytes, quail proepicardia were isolated, cultured and expanded in vitro using a hanging drop system to generate embryonic bodies (EBs) (Appendix A). To promote cartilage differentiation, these EBs were incubated in a standard chondrogenic medium (see material and methods section) (Appendix A).

Twenty-one days after induction, an enrichment on proteoglycans (Figure 2L,M), SOX9-positive nuclei staining (Figure 2N) and COL2A1 deposits (Figure 2O) were observed in the PE-derived EBs.

## 3. Discussion

The presence of discrete masses of cartilage in the heart is a well-known histomorphological trait of many vertebrate species such as the cow [15], the camel [16], the water buffalo [17], the sheep [18] and the otter [19]. In these animals, chondrocytes preferentially differentiate close to the valvular ring at the base of cardiac great vessels [1,3,4].

The possible functions of cardiac cartilage have been extensively discussed, although no clear conclusions are to be found in the literature. Remarkably, in some vertebrate taxa such as bovines, chondrification events were described at perinatal stages in physiological, spontaneous conditions [20]. This is the case of cardiac cartilage development in the chick: while aorta and pulmonary valve chondrocyte clusters (neural crest derived) differentiate at embryonic stages, the new-described PE-derived LOT cluster described in our work differentiates at perinatal stages. These findings suggest a developmental delay in the differentiation of LOT chondrocytes with respect to the aortic and pulmonary ones.

In contrast with the described spontaneous cartilage differentiation in the avian heart, the presence of cartilage in the human heart is regarded as a rare event, and associates with aging and some pathological conditions [21,22,23]. This finding prompted us to test whether anomalous developmental conditions could alter the normal pace of cardiac cartilage differentiation. Our experiments, which combine cell fate tracing methods (quail-to-chick avian chimeras) with controlled damage of the embryonic tissues, revealed an accumulation of EPDCs at the site of injury, as well as the differentiation of some of these cells into cartilage in response to the experimental insult.

The cellular origin of cardiac cartilage has also been extensively discussed, but the current consensus is that cardiac chondrocytes derive from the neural crest [2]. Since the migration of these cells distally to the base of the cardiac outflow tract is considered an infrequent event [9,10], we did seek to study whether an alternative source for cardiac cartilage exists. Our first candidate was the epicardial-derived mesenchyme, that contributes to a large portion of cardiac ventricular interstitium, coronary vessel walls, atrioventricular cardiac valve primordium mesenchyme, and the *annulus fibrosus* [24]. This latter element is in contact with the LOT, a cardiac location relevant to cartilage differentiation. The hypothesis of epicardial lineage cells differentiating into chondrocytes is in accordance with the ability of proepicardial progenitor (proepicardial) cells to differentiate into multiple mesodermal cell types [12] (Figure 3). Cell types differentiating from the proepicardium include endothelial cells [25,26,27,28], smooth muscle cells [29,30,31], cardiac fibroblasts [31,32,33,34,35], adipocytes [36], some cardiomyocytes [37,38,39,40,41,42] and circulating cells [43,44]. Our work is the first one reporting on the chondrogenic potential of proepicardial derived cells both in vivo and in vitro. This discovery adds further evidence to the wide mesodermal/connective tissue potential of epicardial lineage cells (Figure 3).

Such chondrogenic differentiation potential could be interpreted as a particular case of connective tissue differentiation program reactivation in a naïve or non-fully committed mesodermal tissue. Indeed, different mesodermal populations have been reported to display a wide differentiation potential, including cartilage formation, when experimentally manipulated (e.g., when submitted to specific signals or heterotopically transplanted) [45,46].

Taken together, our results confirm that epicardial-derived mesenchymal cells are able to differentiate into cartilage in defined domains of the developing and postnatal avian heart. Our data also suggest that these cells are sensitive and responsive to changes in their milieu. All these findings are relevant to our understanding of cardiac tissue responses to pathologic stimuli and strongly argue in favour of epicardial-derived mesenchyme as a cell source for cell-based reparative therapies, but also as the possible origin of anomalous connective tissue differentiation in the diseased heart.

## 4. Materials and Methods

### 4.1. Avian Embryos

The animals used in our research program were handled in compliance with the international guidelines for animal care and welfare. This work used avian embryos only, which are not regarded as experimental animal models sensu stricto by the European Directive 2010/63/UE, so that no formal ethical approval is required for work on these subjects. Eggs were kept in a rocking incubator at 38 °C. Due to developmental similarities between chick and quail embryos during the first days of incubation [47], both models were staged according to the Hamburger and Hamilton (1951) stages of chick development [48].

### 4.2. Quail to Chick Chimeras

Quail donor embryos were incubated around 60 h until stages 16–17 of development [47], excised and washed in sterile PBS. For proepicardial transplantations, quail proepicardia were carefully dissected using tungsten needles, small iridectomy forceps and scissors and transplanted into prospective pericardial cavity, close to the inner curvature of HH16-17 (60 h of incubation) chick embryo host hearts. For the endocardial chimeras, HH16-17 quail ventricles were isolated and opened in sterile PBS, and then grafted into the prospective pericardial cavity (the quail endocardium facing to the chick myocardium) of HH16-17 chick embryos.

### 4.3. PE Isolation and Culture

PE were isolated from quail embryos as previously described [25]. Two PE per well were cultured in four-well plates (VWR) in DMEM (Gibco, Waltham, MA, USA) supplemented with 10% fetal bovine serum, 2% chick serum (Sigma, St. Louis, MO, USA), 1% L-Glutamine and 1000 IU penicillin/streptomycin (Gibco). After 4 days of incubation at 37 °C and 5% CO_2_, proepicardial cells were progressively expanded up to the third passage. Then, embryoid bodies (EBs) were formed by incubating 100,000 cells in hanging drop culture overnight. After that, each EB was transferred to a well in a low adherent V-bottom microtiter plate (Deltalab) and incubated with the chondrogenic medium (StemPro Chondrogenesis Differentiation Kit, Gibco) for 7 and 21 days.

### 4.4. Samples Treatment

Chick embryos and quail-to-chick chimeras were isolated, washed in PBS and fixed in Methanol:DMSO (4:1) overnight at −20 °C. Adult chick hearts were washed in PBS, dissected and fixed in 10% PFA overnight at room temperature. Embryonic bodies (EBs) derived from PE cultures were fixed in Methanol:DMSO (4:1) for 10 min. All samples were dehydrated in a graded series of ethanol, cleared in butanol, embedded in paraffin (56 °C), sectioned (10 μm) and mounted on microscope slides.

### 4.5. Histochemistry

Mallory’s trichrome staining of paraffin embedded embryonic and adult samples was performed as previously described [49]. Some samples were stained with a commercial Alcian blue solution (Sigma, B8438) for 30 min at room temperature, then washed in distilled water and mounted. Other samples were stained in an aqueous solution of 1.5% Safranin-O (Sigma, S8884), washed in distilled water, counterstained with an ethanolic solution of 0.02% Fast Green FCF (Sigma, F7258), cleared with 1% acetic acid, washed and mounted.

### 4.6. Immunohistochemistry

All samples were dewaxed, rehydrated and washed in PBS. For Col2a1 immunostaining, samples were incubated in 0.25% pepsin (Fluka 77152, in Tris-HCl 10 mM pH2) for 2–4 min at 37 °C. Nonspecific binding sites were blocked in 16% sheep serum, 1% bovine serum albumin, and 0.5% Triton X-100 in Tris-PBS (SBT) for 1 h at room temperature (RT). Avian tissue samples were incubated overnight at 4 °C with anti-COL2A1 primary antibody and 1 h with donkey anti-Mouse IgG AF647 (Jackson, 715-175-150) secondary antibody. All nuclei were counterstained with DAPI (1/2000, Sigma D9542).

For QCPN immunostaining, nuclear epitopes were unmasked with TEG buffer (0.12% trizma base and 0.02% EGTA diluted in distiller water; pH 8.95–9.1) in a pressure cooker for 10 min. Endogenous peroxidase activity was quenched incubating the sections for 30 min in 3% hydrogen peroxide. Endogenous biotin was blocked with avidin/biotin kit (Vector Laboratories, SP2001), and incubated in 16% sheep serum, 1% bovine serum albumin, and 0.5% Triton X-100 in Tris-PBS (SBT) for 1 h at room temperature (RT). Samples were then incubated overnight at 4 °C with QCPN primary antibody, incubated for 1 hr at RT in biotin-conjugated goat anti-mouse IgG (B7264; Sigma-Aldrich, Madrid, Spain), and incubated for 1 h at RT in streptavidin–peroxidase complex (S5512; Sigma-Aldrich). Finally, sections were washed, and peroxidase activity was detected using SIGMAFAST 3,3′-diaminobenzidine tablets (D4293; Sigma-Aldrich). Tissues were counterstained with alcian blue solution (Sigma, B8438) for 30 min.

The collagen type II antibody, developed by Holmdahl, R. and Rubin, K., and the QCPN antibody, developed by Bruce M. and Jean A. Carlson, were obtained from the Developmental Studies Hybridoma Bank, created by the NICHD of the NIH and maintained at The University of Iowa, Department of Biology, Iowa City, IA 52242.

### 4.7. Quantifications

Cell density on cardiac cartilage of chick adult hearts was measured by quantifying cell number and cartilage area with ImageJ (Fiji) software.

### 4.8. Statistics

Statistical analyses were performed by using GraphPad Prism software (GraphPad Software, San Diego CA, USA). t-test with Welch correction were performed in all cases (see figure legends for additional statistical details); *p*-value significance was defined as follows: * *p* < 0.05, ** *p* < 0.01.

## Figures and Tables

**Figure 1 ijms-23-03614-f001:**
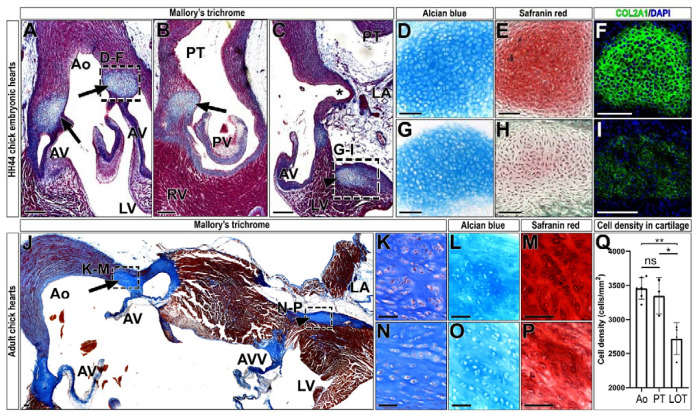
Chick cardiac chondrocyte clusters are ontogenetically heterogeneous. (**A**–**I**) Identification of chondrocyte clusters in HH44 chick embryonic hearts. Mallory’s trichrome staining shows an accumulation of fibrous tissue (blue) in the aortic (**A**) and pulmonary valves (**B**), and in the LOT region (**C**). These clusters were identified by Alcian blue (**D**,**G**), Safranin red (**E**,**H**) and COL2A1 (**F**,**I**) staining. (**J**–**Q**) Identification and characterization of chondrocyte clusters in the adult chick heart. Mallory’s trichrome staining shows cartilage clusters in the aortic valve (arrow in **J**, magnified in **K**) and the LOT region (arrowhead in **J**, magnified in **N**). Aortic valve and LOT chondrocytes were identified by Alcian blue (**L**,**O**) and Safranin (**M**,**P**) staining methods, respectively. Chondrocyte clusters within the aorta and the pulmonary valves showed no significant differences in cell density (cells/mm^2^) (ns *p* = 0.57), while LOT chondrocyte accumulations showed a lower cell density than aorta (** *p* = 0.003) and pulmonary valve (* *p* = 0.03) ones. Asterisk (*) in C points to the left coronary aortic sinus. Ao, aorta; AV, aortic valve; AVV, atrioventricular valve; LA, left atrium; LV, left ventricle; PT, pulmonary trunk; PV, pulmonary valve; RA, right atrium; RV, right ventricle. Scale bars: (**A**–**C**,**J**): 200 µm; (**D**–**I**,**K**–**P**): 50 µm.

**Figure 2 ijms-23-03614-f002:**
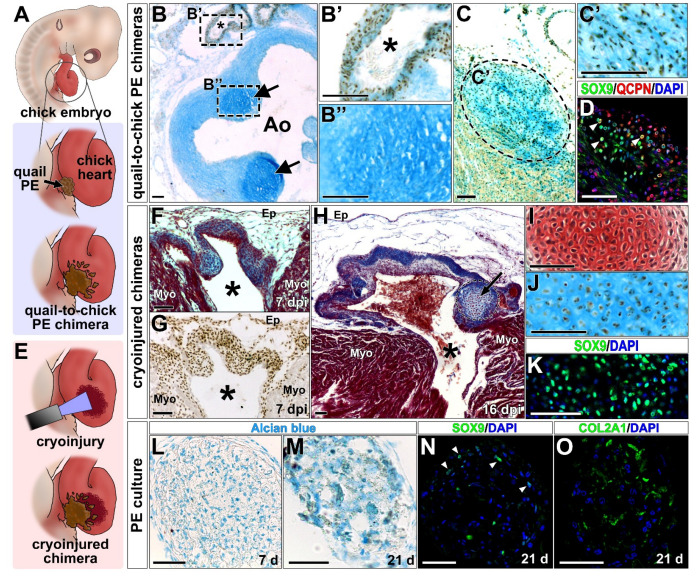
The proepicardium is a source for chondrocytes in the developing heart. (**A**) Quail-to-chick proepicardial chimeras were used to dissect the proepicardial contribution for chondrocytes in vivo. (**B**) In quail to chick PE chimeras, alcian blue and QCPN staining show no contribution of quail cells (QCPN+ cells, black nuclei) to chondrocytes within the aorta or pulmonary valve regions at HH46 stage (arrows; aortic cartilage is magnified in **B’’**). Proepicardial derived cells contribute to coronary vessels close to the embryonic aorta (**B**,**B’**; asterisks point to the lumen of a coronary vessel). (**C**) Mallory’s trichrome staining reveals an accumulation of fibrous tissue (blue) in the LOT region. (**C’**) This cluster shows QCPN positive cells in the alcian blue stained matrix. (**D**) QCPN and SOX9 co-staining (arrowheads) evidence the proepicardial contribution to this cluster. (**E**) Quail-to-chick chimeras were constructed over chick cryodamaged hearts. (**F**,**H**) Mallory’s trichrome of cryoinjured chimeras shows a myocardial discontinuity (asterisk), covered by a fibrotic (blue) region, corresponding to the damage area. (**G**) The fibrotic region is full of QCPN-positive cells at 7 days post-injury (HH35-36). (**H**,**I**) At HH45-46 stage (16 days post-injury), Mallory’s trichrome (**H**, arrow) and Safranin red staining (**I**) reveal chondrocyte clusters within the damage area. (**J**) Alcian blue and QCPN co-staining evidenced the proepicardial contribution to chondrocytes in these clusters. (**K**) Chondrocyte clusters in the damaged region are positive for SOX9 staining. (**L**–**O**) Chondrogenic differentiation is observed in cultured quail PE. (**L**,**M**) Proepicardium-derived embryonic bodies (EBs) show an increase in alcian blue intensity from 7 (**L**) to 21 (**M**) days of chondrogenic induction. (**N**) Nuclear expression of SOX9 is described in these EBs after 21 days of chondrogenic induction (arrowheads). (**O**) These PE-derived EBs also show COL2A1 accumulations. Ao, aorta; EB, embryonic body; Ep, epicardium; Myo, myocardium; PE, proepicardium. Scale bars: 50 µm.

**Figure 3 ijms-23-03614-f003:**
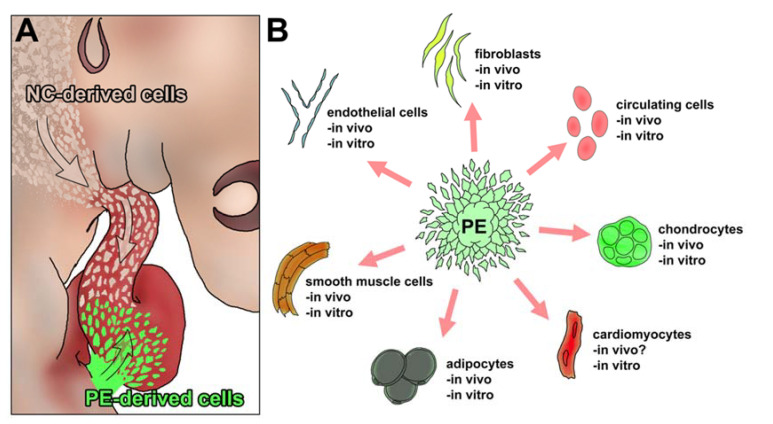
Schematic illustration of the in vivo and in vitro differentiation potential of the proepicardium. (**A**) A model for the integration of different cardiac chondrogenic cell population is shown. (**B**) Schematic illustration showing different reports of in vivo and in vitro differentiation of proepicardial (PE) derived cells into different cell types. These cell types include fibroblasts [31,32,33,34,35], circulating cells [43,44], cardiomyocytes [37,38,39,40,41,42], adipocytes [36], smooth muscle [29,30,31] and endothelial cells [25,26,27,28]. In this publication we report the differentiation potential of the PE both in vivo and in vitro. PE: proepicardium.

## Data Availability

Not applicable.

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
