# Peer review of "In Vivo and In Vitro Cartilage Differentiation from Embryonic Epicardial Progenitor Cells"

_ijms, 2022, doi:10.3390/ijms23073614_

Round 1

Reviewer 1 Report

The study by Palmquist-Gomes et al, analyzes the presence of cartilaginous tissue in the chick heart and provides convincing evidence, obtained from elegant tracing quail-chick experiments, for a specific origin of the chondrogenic cells from the proepicardial cells. The study is clearly written and bear interest for people interested in heart morphogenesis and also in regenerative medicine. However, I think that previous to its acceptance, the authors should analyze whether the structure of the root of the great vessels is similar in quail and chick specimens. This will provide further support to the lineage tracing study.

There are other aspects, which are not strictly required, but which would improve the manuscript. Thus, considering the plasticity of the connective tissues in adult vertebrates to form cartilage under physiological or pathological conditions (see, Montero et al., 2012. Tissue Eng Regen Med. 6:337. doi: 10.1002/term.436), I would suggest the authors to compare the chondrogenic potential and/or the transcriptional profile of skeletal progenitors of the proepicardial cell population with other well-characterized skeletal progenitors (i.e. limb, or branchial arch, mesoderm). For this purpose, cultures at high density (micromass cultures), using different cell concentrations and serum-free medium, might be useful. Alternatively, the authors may consider to extend a bit more the discussion section of the manuscript to comment whether there is a specific commitment of progenitors to form cartilage or if it is a general potential of mesodermal cells.

Author Response

We thank reviewer#1 for the useful comments provided. We detail our specific answers to the points raised below:

  1. The study by Palmquist-Gomes et al, analyzes the presence of cartilaginous tissue in the chick heart and provides convincing evidence, obtained from elegant tracing quail-chick experiments, for a specific origin of the chondrogenic cells from the proepicardial cells. The study is clearly written and bear interest for people interested in heart morphogenesis and also in regenerative medicine. However, I think that previous to its acceptance, the authors should analyze whether the structure of the root of the great vessels is similar in quail and chick specimens. This will provide further support to the lineage tracing study.

We have added a new supplementary figure (Fig. S3) showing the similarities between the root of the aorta and pulmonary trunk in the two avian species used in this study (chicken and quail). We agree this comparison is relevant since we are performing grafting experiments from quail to chick.

  1. There are other aspects, which are not strictly required, but which would improve the manuscript. Thus, considering the plasticity of the connective tissues in adult vertebrates to form cartilage under physiological or pathological conditions (see, Montero et al., 2012. Tissue Eng Regen Med. 6:337. Doi: 10.1002/term.436), I would suggest the authors to compare the chondrogenic potential and/or the transcriptional profile of skeletal progenitors of the proepicardial cell population with other well-characterized skeletal progenitors (i.e. limb, or branchial arch, mesoderm). For this purpose, cultures at high density (micromass cultures), using different cell concentrations and serum-free medium, might be useful. Alternatively, the authors may consider to extend a bit more the discussion section of the manuscript to comment whether there is a specific commitment of progenitors to form cartilage or if it is a general potential of mesodermal cells.

We fully agree the comparative analysis on the chondrogenic potential of proepicardial progenitors is a very interesting topic but, unfortunately it falls beyond the main scope of this study.
In the past, we performed several in vivo experiments in which quail proepicardia were grafted into chick developing limb buds. In all cases, the donor grafts were extruded by the limb mesenchyme, and the chimeras were never successful. We interpreted this as a consequence of adhesive tissue sorting, as the proepicardial tissue is E-Cadherin-positive (Cano et al., 2013, Journal of Developmental Biology 1:3-19) and the limb mesenchyme is mostly N-Cadherin-positive (Hatta et al., 1987, Dev. Biol. 120:215-227). However, this is a research line we are now pursuing to fully unveil the mesodermal differentiation potential of the proepicardium. Our strategy certainly includes transcriptomic approaches, but we cannot yet provide robust preliminary data on this study. Having said this, we also believe that a comment on the differentiation potential of embryonic mesodermal tissues nonetheless could be helpful to the reader. We have thus included in the discussion the following text: “Such chondrocytic differentiation potential could be interpreted as a particular case of connective tissue differentiation program reactivation in a naïve or non-fully committed mesodermal tissue. Indeed, different mesodermal populations have been reported to display a wide differentiation potential, including cartilage formation, when experimentally manipulated (e.g. when submitted to specific signals or heterotopically transplanted) (Schultheiss et al., 1997, Genes. Dev. 11: 451-462; Montero et al., 2012. Tissue Eng Regen Med. 6:337)”.

Reviewer 2 Report

In this manuscript, Palmquist-Gomes and colleagues, approach for the first time, the role of epicardial cells as another source of some of the cartilage tissue found in the embryonic heart.

This is novel as so far, the presence of cartilage tissues in the developing and adult heart, have been linked to neural crest cell derivatives.

Here, the authors claim that at least part of the cartilaginous nodes found in the aortic ring-left cardiac outflow tract domain derive from the epicardial lineage.

The authors used a combination of pure embryonic experiments and in vitro ones. Using quail-chick chimeras to perform in vivo proepicardial cell lineage tracing. They have also convincingly demonstrated, using an in vitro system, the chondrogenic differentiation potential of the proepicardial cells. The manuscript is well written, presenting in an adequate format the hypothesis and the experiments that support the proposed conclusions. The experimental figures are of top quality and the scheme in Figure 3 very helpful.

Author Response

We thank reviewer#2 for the positive comments on our study. No changes in the manuscript were requested by this referee.